Resource

# A comparative study of structural variant calling in WGS from Alzheimer's disease families

John S Malamon[1,*], John J Farrell[2,*], Li Charlie Xia[3,4], Beth A Dombroski[1], Rueben G Das[1], Jessica Way[6], Amanda B Kuzma[1], Otto Valladares[1], Yuk Yee Leung[1], Allison J Scanlon[1], Irving Antonio Barrera Lopez[1], Jack Brehony[1], Kim C Worley[7], Nancy R Zhang[4], Li-San Wang[1], Lindsay A Farrer[2,8,9], Gerard D Schellenberg[1], Wan-Ping Lee[1,*], Badri N Vardarajan[5,*]

Detecting structural variants (SVs) in whole-genome sequencing poses significant challenges. We present a protocol for variant calling, merging, genotyping, sensitivity analysis, and laboratory validation for generating a high-quality SV call set in whole-genome sequencing from the Alzheimer's Disease Sequencing Project comprising 578 individuals from 111 families. Employing two complementary pipelines, Scalpel and Parliament, for SV/indel calling, we assessed sensitivity through sample replicates (N = 9) with in silico variant spike-ins. We developed a novel metric, D-score, to evaluate caller specificity for deletions. The accuracy of deletions was evaluated by Sanger sequencing. We generated a high-quality call set of 152,301 deletions of diverse sizes. Sanger sequencing validated 114 of 146 detected deletions (78.1%). Scalpel excelled in accuracy for deletions ≤100 bp, whereas Parliament was optimal for deletions >900 bp. Overall, 83.0% and 72.5% of calls by Scalpel and Parliament were validated, respectively, including all 11 deletions called by both Parliament and Scalpel between 101 and 900 bp. Our flexible protocol successfully generated a high-quality deletion call set and a truth set of Sanger sequencing–validated deletions with precise breakpoints spanning 1–17,000 bp.

## Introduction

Human genetic variation includes single nucleotide variants (SNVs), small insertions and deletions (indels) less than 50 bp, and structural variants (SVs) greater than 50 bp. SVs can result from deletions, duplications, insertions, and rearrangements that include balanced inversions and translocations or unbalanced repeats, duplications, and deletions resulting in copy-number variation (CNV) (Iafrate et al, 2004; Sebat et al, 2004; Tuzun et al, 2005). SV/indels arise as both single and complex events via germline and somatic mutations (Feuk et al, 2006) and contribute significantly to genetic diversity and to disease susceptibility (Juyal et al, 1996; Ji et al, 2000; Lin et al, 2000; Lupski & Stankiewicz, 2005; Weischenfeldt et al, 2013; Østern et al, 2013; Carvalho & Lupski, 2016).

A variety of SV/indel types and sizes can be detected using high-throughput short-read whole-genome sequencing (WGS). Multiple large-scale SV detection studies have been performed such as the 1000 Genomes Project (Sudmant et al, 2015), the Cancer Genome Atlas Project (Cancer Genome Atlas Research et al, 2013; Fredriksson et al, 2014), Genome of the Netherlands (Genome of the Netherlands Consortium, 2014), the UK 10K Project (UK10K Consortium et al, 2015), gnomAD (Collins et al, 2020), and CCDG (Abel et al, 2020). However, SV/indel calling using short-read sequence data continues to be challenging. Multiple algorithms and programs (e.g., Breakdancer, CNVnator, DELLY, Genome Analysis Toolkit [GATK: 3.2] Haplotype Caller, Lumpy, Pindel, Scalpel, and SWAN) (Chen et al, 2009; Ye et al, 2009; McKenna et al, 2010; Abyzov et al, 2011; Rausch et al, 2012; Layer et al, 2014; Narzisi et al, 2014; Xia et al, 2016) are available, but many factors continue to hinder accurate and comprehensive identification of SV/indels in sequence data. These confounding factors include complex sequence structure, variability in read depth and coverage across the genome, sequencing bias and artifacts, biological contamination, and mapping and alignment errors or artifacts. Also, computational demands can limit the use of some SV/indel calling programs. Furthermore, SV/indel calling in large samples lacks standards for calling procedures, call set merging, and quality control (QC). These challenges become even more daunting when merging SV/indel

[1]Department of Pathology and Laboratory Medicine, Penn Neurodegeneration Genomics Center, University of Pennsylvania Perelman School of Medicine, Philadelphia, PA, USA [2]Biomedical Genetics Section, Department of Medicine, Boston University School of Medicine, Boston University, Boston, MA, USA [3]Division of Oncology, Department of Medicine, Stanford University School of Medicine, Stanford, CA, USA [4]Department of Statistics, The Wharton School, University of Pennsylvania, Philadelphia, PA, USA [5]Gertrude H. Sergievsky Center and Taub Institute of Aging Brain, Department of Neurology, Columbia University Medical Center, New York, NY, USA [6]Broad Institute, Massachusetts Institute of Technology, Cambridge, MA, USA [7]Human Genome Sequencing Center, and Department of Molecular and Human Genetics, Baylor College of Medicine, Houston, TX, USA [8]Departments of Neurology and Ophthalmology, Boston University School of Medicine, Boston University, Boston, MA, USA [9]Departments of Epidemiology and Biostatistics, Boston University School of Public Health, Boston, MA, USA

Correspondence: wan-ping.lee@pennmedicine.upenn.edu; bnv2103@cumc.columbia.edu
*John S Malamon, John J Farrell, Wan-Ping Lee, and Badri N Vardarajan contributed equally to this work

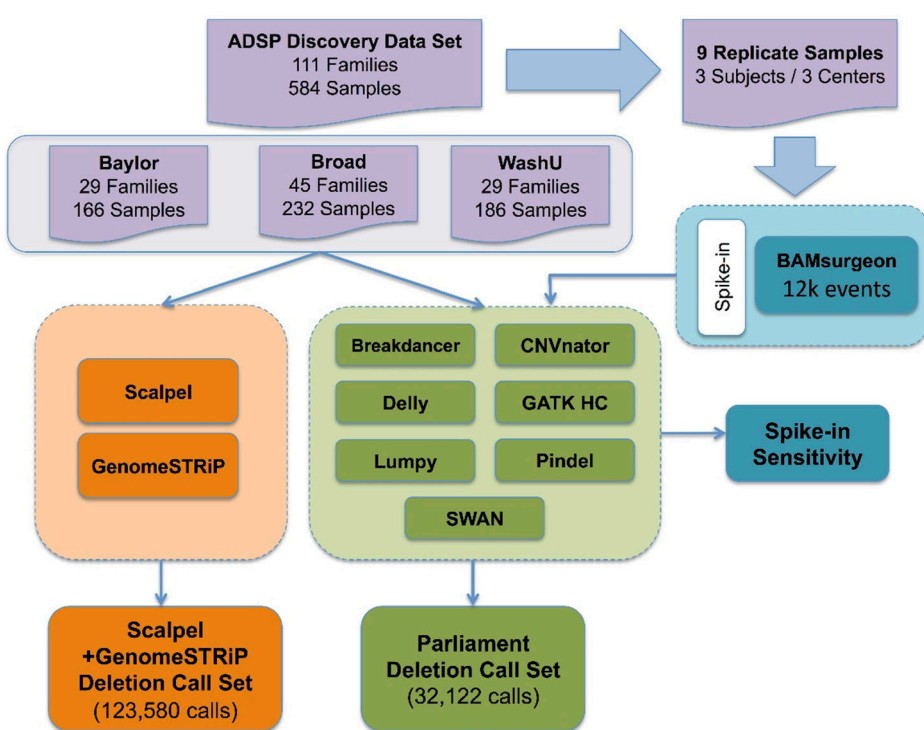

**Figure 1. Overview of Alzheimer's Disease Sequencing Project's SV/indel calling and analysis pipeline.**
Two parallel pipelines, Scalpel + GenomeSTRiP (orange) and Parliament (green), were combined to perform SV/indel call merging, QC, genotyping, and reassembly for 584 samples from three sequencing centers. Nine replicated samples were used to measure individual SV/indel caller sensitivity via variant spike-in studies.

calls from samples sequenced at multiple centers that use different sequencing library designs and protocols. The quality and characteristics of sequence data may vary considerably among samples within and across centers and can affect SV/indel calling sensitivity and specificity (Guo et al, 2014). We present results from analyses of WGS data generated by the Alzheimer's Disease Sequencing Project (ADSP) for 578 members of 111 families. We focused on deletions because deletion detection is more reliable than other SV types. In addition, we spiked in structural variants in existing sequences for benchmarking and evaluating the performance of several of these tools and pipelines. Spiking in known structural variants allows for a systematic evaluation of a tool's sensitivity and specificity. Spiking in variants of different sizes and complexities allowed us to benchmark tools across a spectrum of genomic alterations, providing a more comprehensive evaluation. In addition, it also facilitated the evaluation of a tool's ability to distinguish true variants from background noise.

Our work showed that no single caller can accurately detect a broad range of deletion sizes. We developed two systematic approaches for evaluating the sensitivity and specificity of different callers and deletions identified from data generated by different platforms and sequencing centers. Finally, we validated a comprehensive strategy for calling, merging, QC, and genotyping deletions that had high sensitivity and minimized false-positive calls.

# Results

We generated deletion calls for the ADSP discovery phase WGS using eight different programs (GATK Haplotype Caller, Scalpel,

Breakdancer, CNVnator, Lumpy, Pindel, Swan, and DELLY) (Methods: Deletion variant calling protocol, Fig 1). These programs use different sequence features and analyze different event sizes (Tables 1, 2, and S1). To determine the properties of the data generated by each program, we systematically evaluated sensitivity and specificity. Because the sequence data were generated at three different Large-Scale Sequencing and Analysis Center (LSAC) sites using libraries with different characteristics (Methods: Subjects and generation of WGS data), we evaluated data from each site. We also benchmarked the computational resources needed for each program.

## Sensitivity analysis

Sensitivity was evaluated by inserting deletions and insertions into WGS data generated at each LSAC (Methods: Sensitivity analysis using simulated spike-in data). Sensitivity for detecting the inserted deletions varied among callers and, to a lesser extent, the source of the sequence data, and was dependent on the size of the deletion (Fig 2). For short deletions (30–500 bp), Scalpel showed the best sensitivity (~85%) and was closely followed by Pindel. Pindel showed good sensitivity up to 1,000 bp. GATK Haplotype Caller showed a sensitivity of ~75% for events up to 100 bp but fell off rapidly above this size range. For larger events, Lumpy and SWAN both showed good performance up to 5,000 bp. DELLY showed reasonable sensitivity in the 500- to 5,000-bp range, but when compared to other programs, its results were more influenced by the source of the data. For example, DELLY had lower sensitivity when calling genomes sequenced by BCM in the 200- to 500-bp bin as compared to those from WashU and BI. Overall, SWAN was the

**Table 1. Overview of SV/indel callers evaluated.**

| Caller | Software version | Sequence feature | Calling range (bp) | Exact genotype | Precise breakpoint |
|---|---|---|---|---|---|
| Breakdancer | 1.1 | RP | 1–10,000 | Yes | No |
| CNVnator | 0.3 | RD | 200–10,000 | No | No |
| DELLY | 0.5.6 | RP and SR | 15–10,000 | Yes | Partial |
| GATK HC | 3.2 | RP | 2–300 | Yes | Yes |
| LUMPY | 0.2.10 | RP, SR, and RD | 1–10,000 | No | No |
| PINDEL | 0.2.5a3 | RP and SR | 1–10,000 | Yes | No |
| Scalpel | 0.5.3 | AS | 1–1,000 | Yes | Yes |
| SWAN | 0.3.0 | RP, SR, and SC | 1–10,000 | No | Partial |

Column 1 provides the caller evaluated. The second column provides the software version used for each caller. "Sequence Feature" provides the method used to determine events such as read-pair (RP), split-read (SR), read-depth (RD), soft-clip (SC), and local assembly (AS). Columns 5 and 6 provide whether each caller supplies precise genotypes and breakpoints.

most sensitive caller across all sizes and sequencing centers, perhaps because it accounts for various sequencing characteristics such as multiple insert-size libraries and soft-clipped reads (Xia et al, 2016; Zhang et al, 2016). CNVnator and Breakdancer showed worrisome sensitivity for all size ranges. Our results show that sensitivity varies considerably among callers and for different size ranges but is relatively insensitive to the sequencing site.

### Specificity analysis

We assessed caller specificity using the D-score method (Methods: D-score: a metric for evaluating SV/indel caller specificity in family studies). LUMPY was the best-performing program with D-scores between 5 and 10 for deletions from 30 to 10,000 bp (Fig 3). The results were independent of the sequencing center. Scalpel also yielded highly specific calls, particularly in the 200- to 1,000-bp range with D-scores ranging from 5 to 8. The median D-scores for deletion calls from SWAN, Pindel, and Breakdancer were between 3 and 5, but the results were dependent on the sequencing center. Other programs yielded calls with lower specificity that were greatly influenced by the sequence source. We also applied the kinship coefficient to evaluate and calibrate the quality of deletion calls and measure the impact of QC steps on call specificity (Fig 4A). Before data cleaning, the kinship coefficient was much greater than the expected value of 0.25 in siblings for events ranging from 21 to 350 bp, suggesting that that Scalpel is overcalling variants in this size range. False positives often occur because of mapping issues with the allele frequencies approaching 50%. This results in higher-than-expected heterozygous genotypes under the Hardy–Weinberg equilibrium. After removing deletions showing excess heterozygosity, the kinship coefficient of the Scalpel genotypes approached 0.25 for all deletion sizes (Fig 4A). Comparison of kinship coefficient metrics also showed that the quality of GATK Haplotype calls decreased as the deletion size increased and the coefficient was 0 for deletions ≥50 bp (Fig 4B). In contrast, a kinship coefficient of 0.25 was maintained for Scalpel calls for deletion sizes between 20 and 400 bp, showing that the Scalpel calls are more reliable in this size range. This work shows that the specificity of calls from different programs varies depending on the size of the event detected and can be influenced by the source of the sequence data.

**Table 2. Total calls by eight SV/indel callers.**

| Caller | Number of calls |
|---|---|
| Breakdancer | 3,484,082 |
| CNVnator | 2,572,070 |
| DELLY | 1,685,852 |
| GATK Haplotype Caller | 232,366 |
| LUMPY | 1,003,953 |
| PINDEL | 2,613,604 |
| SWAN | 1,945,009 |
| Scalpel | 1,441,659 |
| Total | 14,709,212 |

Number of pre-QC calls for all eight callers.

### Assessment of SV/indel caller computational requirements

We measured computational performance metrics for seven of the eight callers used in this study (Methods: Computational performance of SV/indel callers, Fig 5, Table S1). Scalpel was excluded from performance benchmarking because of its extreme central processing unit (CPU) demands and total runtime. To generate these benchmark metrics, we processed 10 BAM files (mean size of 209.05 MB) from the ADSP's discovery (disc) phase and 10 BAM files (mean size of 54.58 MB) from the discovery extension (disc + ext) phase. Among the tested callers, SWAN had the highest memory demands and required more than 10 times greater runtime compared with other programs. Breakdancer was the second longest running SV caller evaluated. DELLY, Lumpy, GATK, and SWAN all had similar CPU demands. Although Scalpel and SWAN ranked high in terms of sensitivity and specificity, the runtime computational requirements preclude the use of these programs on large datasets.

### Generating an ADSP deletion call set

All 584 samples were called in parallel via two independent production pipelines, Scalpel + GenomeSTRiP and the Parliament

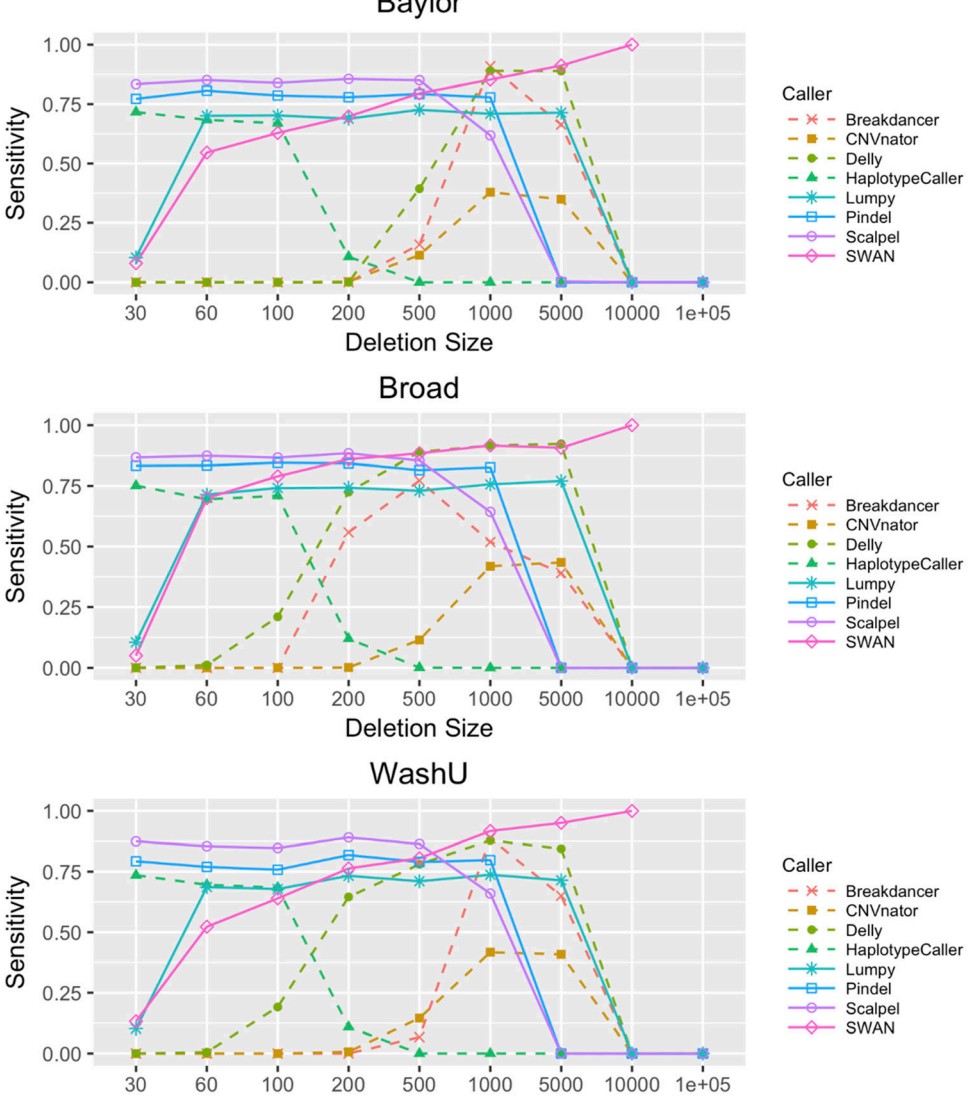

**Figure 2.   SV/indel caller sensitivity stratified by sequencing center and caller.** Sensitivity rates were derived for all eight callers using the in silico variant spike-in on nine sample replicates. Sensitivity is provided for all three centers (Baylor, Broad, and WashU). Biological replicates are three individuals in one family that were sequenced at the three centers. Sensitivity rates are provided across a large range of event sizes (30 bp–10 kb). Sensitivity rates are largely consistent across centers.

Toolkit (Fig 1). Given that the sensitivity and quality of the GATK Haplotype Caller dropped off significantly with deletions of size greater than 20 bp, the pipelines focused on deletions greater than that size range. Localized assembly and breakpoint refinement on gapped alignments were performed with Scalpel to increase the calling accuracy of deletions as large as ~900 bp. Of the 123,581 deletions detected by Scalpel and genotyped with GenomeSTRiP, 100,678 sites remained after the removal of deletions with excess heterozygotes (N = 17,286), homozygous reference (N = 5,014), and call rates less than 90% (N = 603). The number of deletions called dropped off exponentially as the deletion size increased except for a spike in the number of deletions related to Alu retrotransposons (Fig 6). The frequency of these events peaked around 350 bp, which corresponds to the lengths of most Alu transposons, and this size distribution is expected and consistent with that observed in other studies (Collins et al, 2020). The Parliament pipeline genotyped more than 14 million SVs from the eight callers listed in Table 2. The

mean number of calls per program was slightly greater than 1.8 million. Because of computational requirements, the sites genotyped were limited to those greater than or equal to 100 bp. A total of 32,122 calls remained post-QC, and the size distribution of these calls shows the Alu peak at ~350 bp (Fig 7A and B). The distribution of functional annotations of these variants is shown in Table 3. A comparison of the deletion calls generated by the two pipelines in the size ranges that overlapped (100–900 bp) identified 3,401 deletions (mean size = 330 bp, range 207–620 bp) that shared a base location for at least one breakpoint (Fig S1) in the size bin with deletions common to both callers.

### Laboratory validation of deletion calls

To validate deletion calls, we performed Sanger sequencing on putative deletions (Methods: Laboratory validation of deletion calls). We sequenced 106 deletions called by Scalpel ranging in size

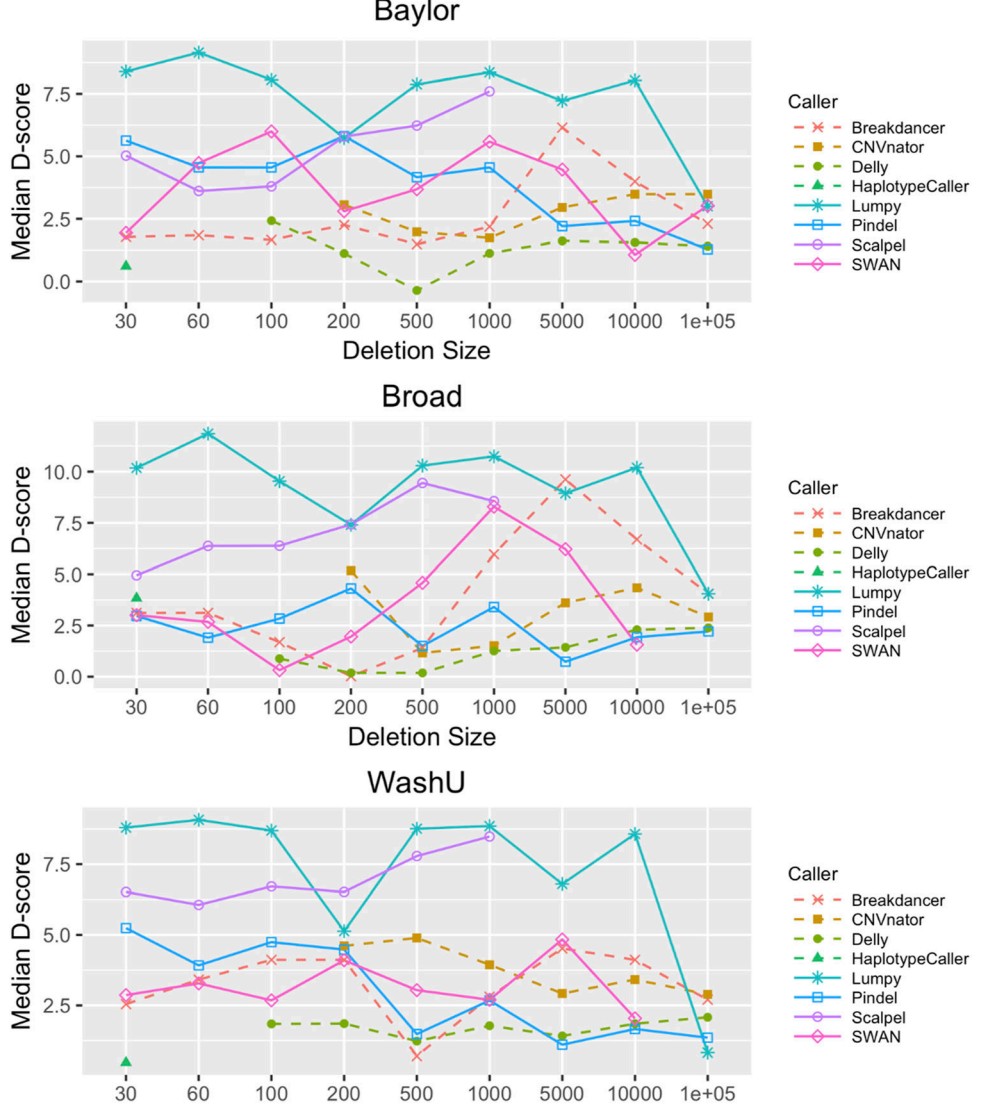

**Figure 3. SV/indel caller specificity using the D-score stratified by sequencing center and caller.**
Specificity rates are provided for all eight callers from 30 bp to 10 kbp using our D-score method. D-scores were calculated for each of the three sequencing centers (Baylor, Broad, and WashU). D-scores are quite consistent across centers.

from 2 to 900 bp (Tables 4 and S3). When smaller deletions were randomly selected, 87.5% of events between 2 and 100 bp were validated by Sanger sequencing (100% of the events under 20 bp and 80% of events between 80 and 100 bp were confirmed by Sanger sequencing). For loss-of-function deletions and those near AD genes (±500 kb, Table S2) in this size range, slightly higher validation rates were observed (average 93% and 95%, respectively). For randomly selected large events (between 101 and 900 bp), the validation rate fell to 17%. This size range includes several transposable elements (e.g., Alu) in the genome that are susceptible to higher false-positive rates in SV calling across most calling algorithms. Although Scalpel detected deletions up to 900 bp, we found the spiked-in sensitivity drops off at 500 bp (Fig 2) as Scalpel's window size was set to 600 bp. For many of the non-validated deletions (Table 4), the sequencing did find an alternate SV that was not a deletion (e.g., repetitive low complexity regions [LCR], ALU insertion). However, when large SV/indel calls were

prescreened to remove deletion sequences found at multiple regions of the genome, the validation rate increased to 50%. Deletions near AD genes and LOF variants had a higher validation rate (83% and 75%, respectively).

For Parliament pipeline calls, 20% of randomly selected deletions in the 101- to 900-size range could be validated. For Parliament calls near AD genes and LOF variants, calls were validated at a higher rate (33% and 83%, respectively). For larger SV calls, the validation rate ranged from 73 to 83%. When we examined calls made by both Parliament and Scalpel, all deletions tested (n = 11) could be validated. The mean D-score for validated deletions (8.12, sd = 10.98, n = 114) was significantly greater than the mean for deletions that were not validated (2.52, sd = 4.98, n = 19, $P$ = 0.0075). This Sanger sequencing validation of deletions demonstrates that the variants called by Scalpel, particularly within the 2- to 100-bp size range, are highly reliable and are suitable for genetic association studies.

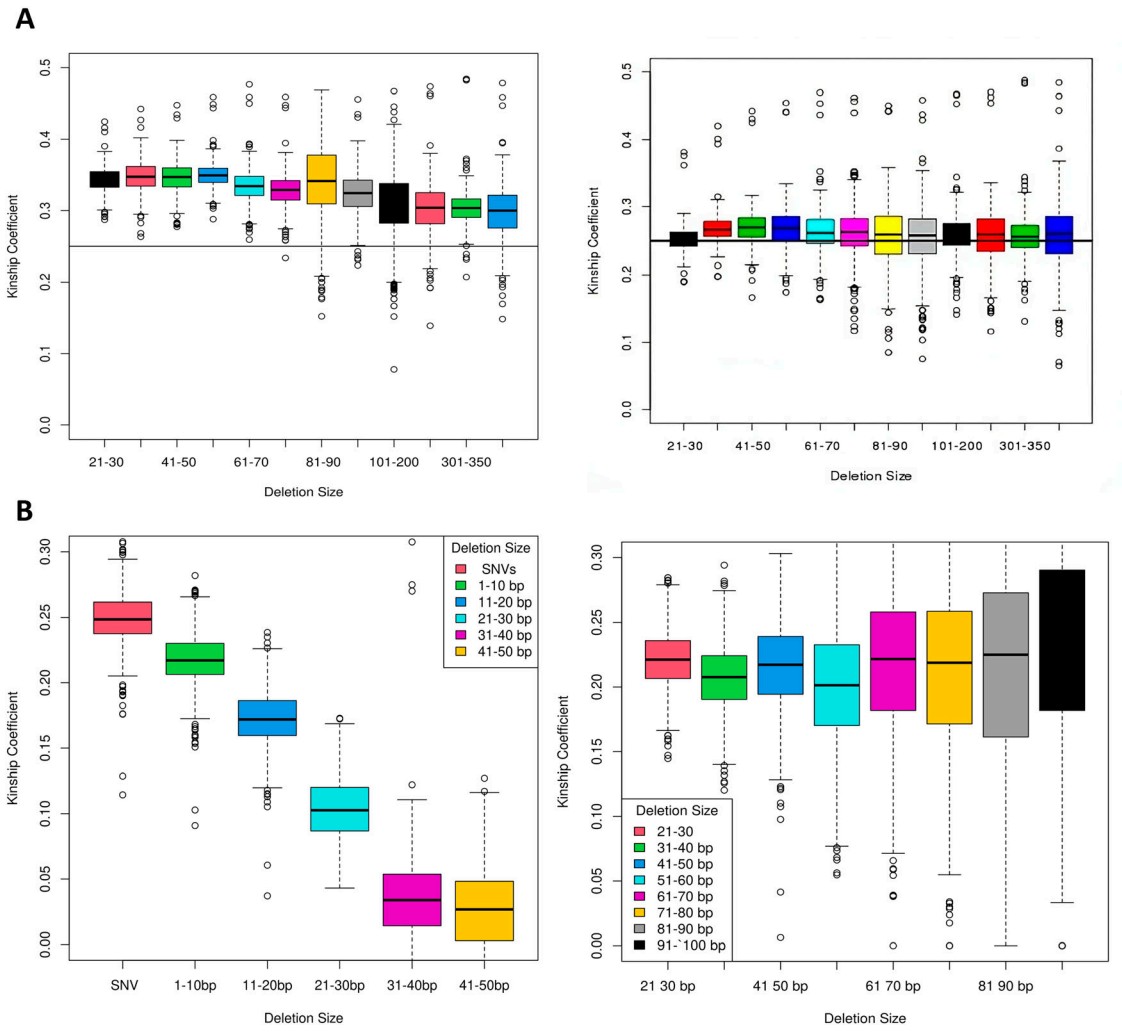

**Figure 4.  Kinship coefficient by deletion size.**
**(A)** pre- and post-QC. Kinship coefficients for pre- (left) and post-QC (right) calls ranging from 20 to 400 bp were calculated for all sibling pairs. QC filtering to reduce excess heterozygosity resulted in coefficients that approximated the expected value of 0.25. **(B)** Kinship coefficients for GATK Haplotype Caller with VQSR using the best practices Mills indel training set (left) and GATK Haplotype Caller with an improved indel training set (right). Kinship coefficient is 0.25 for SNVs called by the GATK Haplotype Caller but declines progressively to 0 with increasing deletion sizes. In contrast, the coefficient approximated 0.25 for the GATK Haplotype calls across all SV/indel bin sizes when using an improved training set for VQSR step that includes deletions across the range of sizes.

## Deletions within/near AD genes

To detect possible AD-associated pathogenic variants, we looked for deletions in a ±500-kb window bracketing candidate AD genes, focusing on deletions in gene functional units (coding regions, 5′ and 3′UTRs, promoters, and splice junctions). This window was selected to capture genes regulated by cis-acting elements impacted by peak GWAS variants that influence the expression of causal AD genes. We identified deletions in the vicinity of 24 AD candidate genes (Table S4) that could be validated by Sanger sequencing. One pathogenic deletion identified using Scalpel was a 44-bp deletion in exon 14 of *ABCA7* (rs142076058, p.Arg578 fs). Subsequent work in a larger sample showed that the deletion was associated with AD in African American populations (Cukier et al, 2016). For the remaining confirmed SVs, we tested the segregation

of the SVs in the families by requiring that at least 75% of the patients with LOAD and WGS data in the families were carriers. We found segregation in six SVs in at least one family near *IQCK*, *FBXL7*, *INPP5D*, *SPDYE3*, and *SERPINB1* (Table S5). A 21-bp coding deletion was identified (rs527464858) in *GIGYF2*, a gene that encodes GRB10-interacting GYF protein 2. This protein regulates tyrosine kinase receptor signaling. The *GIGYF2* deletion is in an imperfect "CAG" repeat sequence and is ~270 kb from rs10933431, the top SNV for *INPP5D* ($P = 3.4 × 10^{-9}$, OR = 0.91, CI: 0.88–0.97) (Kunkle et al, 2019). This deletion was observed in 46 cases and 3 controls in both NHW and CH populations. Note that in our study, there were more cases (n = 498) than controls (n = 86) and some of these subjects are related (n = 111 families). This deletion was observed in 10 CH and 13 NHW families. Co-segregation showed that the variant segregated with the AD status in three NHW families and one Hispanic family.

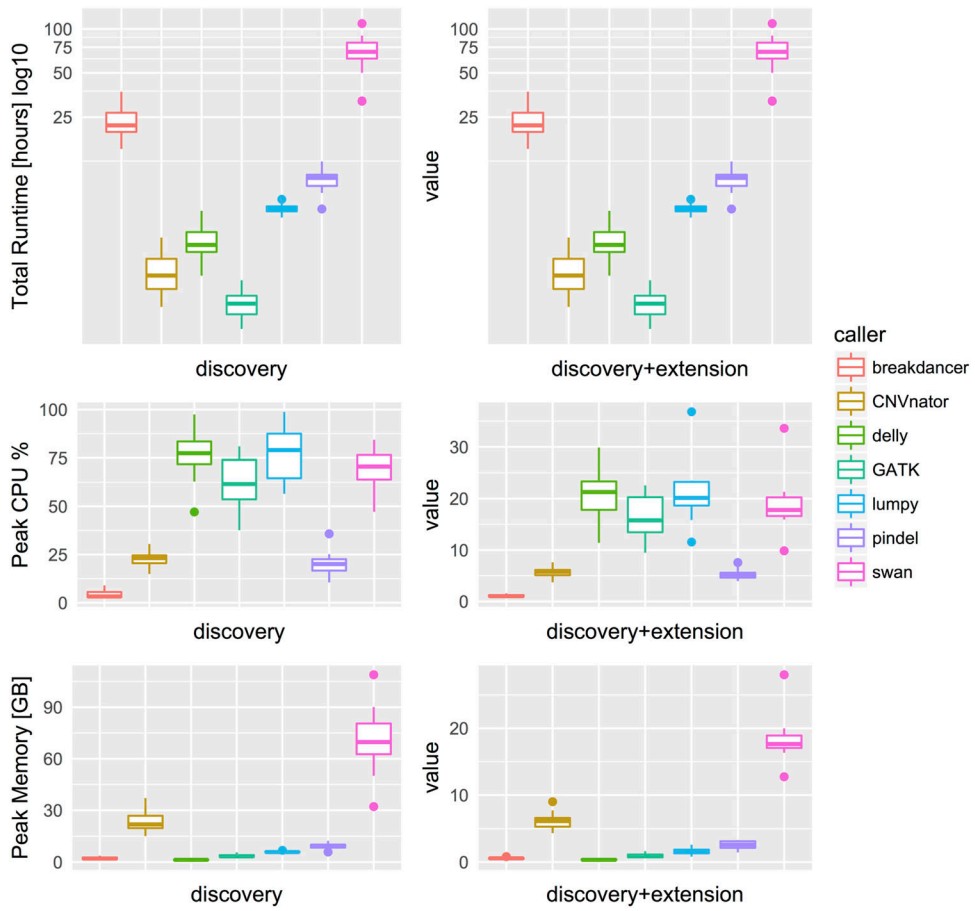

**Figure 5. Three performance metrics for seven SV/indel callers.**
top row provides the total runtime in hours, the middle row provides peak central processing unit percentage, and the bottom row provides peak memory in gigabytes for 10 discovery phase (left column) and 10 discovery extension phase (right column) samples.

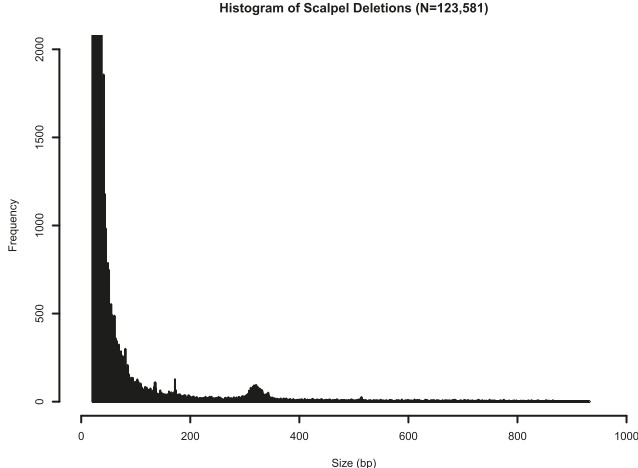

**Figure 6. Histogram of Scalpel deletions by size.**
All Scalpel deletions (N = 123,581), ranging from 20 to 900 bp. The y-axis is truncated at 2,000 calls. The Alu peak is seen near 350 bp.

A number of studies have found variants in *GIGYF2* potentially associated with an autosomal dominant form of Parkinson's disease (Lautier et al, 2008; Ruiz-Martinez et al, 2015; Cristina et al, 2020), particularly in European populations but not in Asian cohorts (Lautier et al, 2008; Ruiz-Martinez et al, 2015; Zhang et al, 2015; Cristina et al, 2020). Although several SNVs in *GIGYF2* may be associated with PD, most studies have not confirmed an association between this gene and PD (Ruiz-Martinez et al, 2015), and a large meta-analysis did not find that PD was associated with the poly-Q region deletion described here (Zhang et al, 2015).

## Discussion

We developed novel approaches for detecting deletions and evaluating sensitivity, specificity, and validity. These methods were applied to WGS data obtained from 578 participants of the ADSP. We evaluated eight SV/indel callers on data generated at three sequencing centers, each of which generated sequence libraries using different protocols. Although sequencing library heterogeneity did not appreciably influence results obtained with most programs, call validity (deletion detection by an orthogonal method) varied by size and calling program. Our results revealed that no single calling program could reliably and accurately detect deletions in all size ranges. Ultimately, we effectively detected and genotyped deletions in the WGS dataset using a combination of SV and indel callers, applying several QC filters, and validating calls by Sanger sequencing.

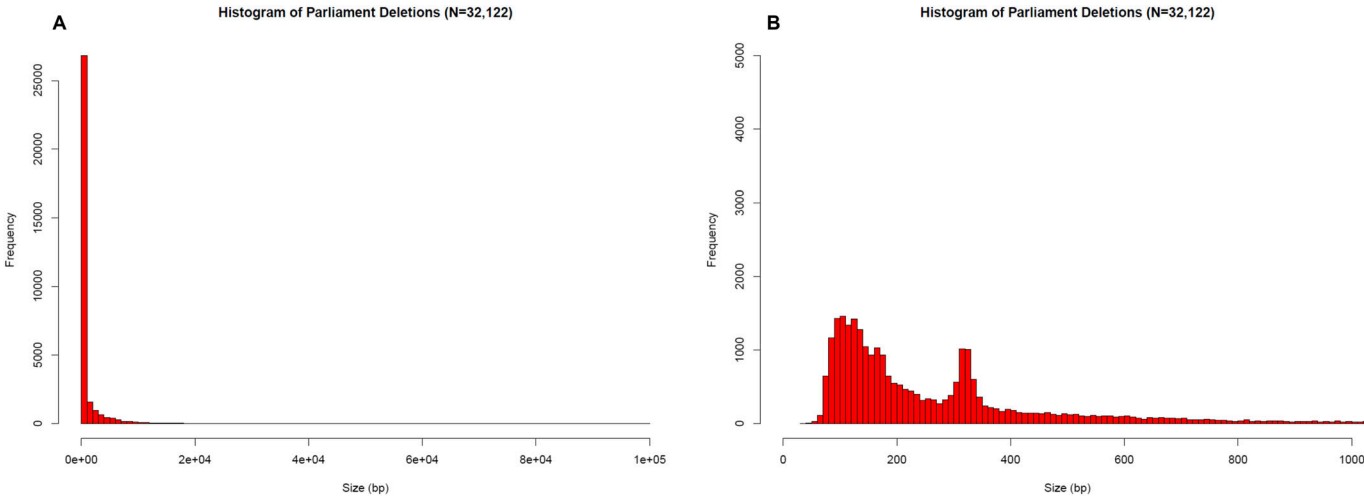

**Figure 7. Histograms of Parliament deletion frequencies by size.**
**(A)** Histogram of Parliament deletions (N = 32,122) ranging from 20 to 1,000 bp. **(B)** Full histogram of all Parliament calls (N = 32,122) ranging from 1 to 10,000 bp. The Alu peak is seen at ~350 bp.

**Table 3. SnpEff functional annotation categories for Scalpel and Parliament calls.**

| Functional annotation term | Scalpel | Parliament | Parliament + Scalpel |
|---|---|---|---|
| Intergenic | 59,595 | 15,238 | 1,900 |
| Coding | 51,849 | 14,724 | 28 |
| Splice site | 339 | 959 | 0 |
| Intronic | 492 | 151 | 1,475 |
| 5'UTR | 201 | 113 | 0 |
| 3'UTR | 739 | 201 | 0 |
| Other | 10,857 | 887 | 0 |
| Totals | 124,072 | 32,273 | 3,403 |

Breakdown of genomic functional annotation terms provided by SnpEff. There are slightly more annotation terms than loci as some loci overlap more than one region.

We evaluated the sensitivity of multiple SV/indel callers by in silico insertion of deletions and insertions into ADSP biological replicate sequence data. This simulation exercise suggests that Scalpel has the highest sensitivity for deletions in the 30- to 500-bp range. Scalpel's sensitivity performance was closely matched by Pindel and the GATK Haplotype Caller, but the latter only for smaller events. Also, the specificity of calls made by Pindel was much less than Scalpel because of the excess number of events called by this program. We measured the specificity of the SV/indel programs using the D-score, a measure that compares deletion sharing between related and unrelated individuals, and the kinship coefficient, which allows a comparison of the observed number of deletion calls with the number of expected calls among individuals with a defined degree of relationship. Scalpel and Lumpy showed the best specificity across a broad size range from 30 to 1,000 bp and were relatively insensitive to sequence library differences. In contrast, the output from other callers was more sensitive to the source of the sequence data.

We developed a comprehensive pipeline for calling, merging, QC, genotyping, and the breakpoint refinement of deletions using Scalpel and GenomeSTRiP (Fig 1). As expected, the most common deletions were small and we observed an excess of deletions of ~350 bp in length, many of which are likely Alu repeat sequences (Figs 5 and 6). For the size bin of 20–100 bp, Sanger sequencing validated more than 87.5% of randomly selected deletions and 90.1% of all deletions (random, near AD genes, LOF variants) (Table 4). This size bracket included 82,180 deletions and accounted for 88.7% of all deletions detected by Scalpel (n = 92,659 total deletions, Table S6). In addition, the Scalpel dataset had a kinship coefficient near the expected value of 0.25 for siblings after the removal of sites with excess heterozygosity.

Our study has several noteworthy strengths. First, we developed a method for evaluating deletion specificity in family-based studies (D-score). This allowed us to directly compare different methods of deletion calling directly using study sequence data. Also, the D-score can be used to prioritize SVs for targeted validation. Second, we used the kinship coefficient metric as a method to measure the overall quality of the call set genotypes and evaluate quality control measures applied to family-based data. Third, we generated spiked-in datasets that allowed for the evaluation of sensitivity in the

**Table 4.  Deletion validation results.**

| Selection method | Workflow | Size bin (bp) | Sequenced[a] | Validated[b] | Failed[c] | Alternate event[d] | No PCR product[e] | Percent validated | |
|---|---|---|---|---|---|---|---|---|---|
| Near AD genes[f] | Scalpel | 20–50 | 12 | 11 | 0 | 1 | 1 | 92% | 90% |
| | | 51–100 | 3 | 3 | 0 | 0 | 0 | 100% | |
| | | 101–800 | 6 | 5 | 1 | 0 | 0 | 83% | |
| | Parliament | 101–900 | 6 | 2 | 1 | 3 | 0 | 33% | 59% |
| | | 1,001–17,000 | 11 | 8 | 3 | — | 0 | 73% | |
| SnpEff LOF | Scalpel | 20–50 | 13 | 13 | 0 | 0 | 0 | 100% | 90% |
| | | 51–100 | 3 | 2 | 1 | 0 | 0 | 67% | |
| | | 101–400 | 4 | 3 | 1 | 0 | 1 | 75% | |
| | Parliament | 100–200 | 3 | 2 | 1 | — | — | 67% | 83% |
| | | 201–400 | 6 | 6 | — | — | 1 | 100% | |
| | | 501–900 | 3 | 2 | 1 | — | 1 | 67% | |
| Anywhere in the genome | Scalpel | 2–19 | 11 | 11 | 0 | 0 | 2 | 100% | 78% |
| | | 20–40 | 6 | 5 | 0 | 1 | 1 | 83% | |
| | | 41–60 | 6 | 5 | 1 | 0 | 2 | 83% | |
| | | 61–80 | 7 | 6 | 1 | 0 | 2 | 86% | |
| | | 81–100 | 10 | 8 | 1 | 1 | 1 | 80% | |
| | | 101–900 | 6 | 1 | 3 | 2 | 0 | 17% | |
| | Parliament | 101–900 | 5 | 1 | 1 | 3 | 4 | 20% | 55% |
| | | 900–1,000 | 6 | 5 | 1 | 0 | 1 | 83% | |
| Cleaned[g] | Scalpel | 101–900 (Cleaned) | 8 | 4 | 2 | 2 | 0 | 50% | 50% |
| In common[h] | Scalpel and Parliament | 101–900 | 11 | 11 | 0 | 0 | 0 | 100% | 100% |
| Totals[i] | | | **146** | **114** | **19** | **13** | **17** | | **78%** |

[a]"Sequenced" are deletions where PCR products were produced that could be sequenced.
[b]"Validated" is the number of deletions where Sanger sequencing yielded the predicted deletion.
[c]"Failed" is the number of confirmed false-positive calls.
[d]"Alternate Events" indicates that a deletion other than the predicted event was observed.
[e]"No PCR Product" is the number of events that could not be amplified and thus could not be tested.
[f]The AD gene list used is in Table S2. Deletions tested were within ±500-kb bp of the target gene.
[g]"Cleaned" indicates that the BLAT was used to exclude events that mapped to multiple places in the genome.
[h]"In common" was randomly selected from a list of identical deletions called by both the Scalpel and Parliament pipelines.
[i]"Totals"—for total Scalpel and Parliament, calls "in common" events were included in the final total for each pipeline.

sequence data used in this study. Fourth, an orthogonal method (Sanger sequencing) was used to validate candidate deletions and to identify the characteristics of true calls. Fifth, the high-quality deletion calls from Scalpel, particularly those under 100 bp, can be used as a gold standard for comparison with calls from other programs that are computationally less intensive. Sixth, we cataloged deletion sites with precise breakpoints that can be directly geno-typed in WGS CRAMs using other genotyping tools such as Graphtyper and Paragraph. Last, we detected a deletion in *ABCA7* that was subsequently shown to be pathogenic. This illustrates the validity of our approach to identifying AD-related deletions.

Our conclusions and recommendations for deletion calling have some limitations. Although the D-score and kinship coefficient are useful specificity measures, they require family-based data. Also, because the D-score method relies on a comparison of the deletion frequency in the general population (i.e., unrelated individuals) versus related individuals, it does not perform well for deletions that are very rare (less than 20 instances in a dataset) or very common with allele frequencies approaching 50%. A minimum of two SVs are needed to compute a D-score. In both cases, the resulting D-score will be close to zero. Second, computational requirements need to be considered. Scalpel, while yielding high-quality calls, is not practical when applied to WGS datasets containing more than a few thousand subjects because this program is computationally intensive. However, the Scalpel calls generated here can be used as a benchmark for evaluating the sensitivity and specificity of other programs such as more recent versions of GATK Haplotype Caller (unpublished data). The utility of callers with longer runtimes can be improved by splitting larger chromosomes and processing them in parallel. However, the cost of using some

programs such as Scalpel and SWAN may be prohibitive when applied to datasets much larger than the one used in this study. Another limitation of our study is that for associations of deletions with AD, our study is underpowered. Thus, we can nominate deletions as candidate pathogenic variants (e.g., Table S4) but will need larger follow-up studies to confirm true associations (e.g., the *ABCA7* deletion). Finally, we only evaluated deletions in this study because of the poor performance of the callers used to detecting insertions and other types of events. Future studies will use other programs that better detect insertions, rearrangements, and copy-number changes.

Findings from this study have multiple, important implications. Small deletions represent a substantial portion of genetic variation (1000 Genomes Project Consortium et al, 2015; Collins et al, 2020). Larger deletions are rarer and account for a small fraction of total genetic variability but are more likely to be deleterious because they may alter large portions of one or more genes. Given the challenges of accurate SV/indel detection and genotyping, SV/indels larger than a few base pairs are typically not included in genetic association studies. Accurately called and genotyped indels/SVs can increase the scope of both hypothesis-driven and genome-wide association studies. Moreover, similar to SNVs, SV/indels in the context of a large WGS or WES dataset can be imputed reliably into GWAS datasets derived from SNP arrays. Studies of SV/indels in the future will likely increase and improve our understanding of the genetic architecture of many diseases as more reliable and efficient calling algorithms are developed and validated.

# Materials and Methods

## Subjects and generation of WGS data

WGS data were obtained from the ADSP, a collaboration between the National Institute on Aging (NIA), the National Human Genome Research Institute (NHGRI), and the Alzheimer's disease research community (Beecham et al, 2017). Details of subject selection and WGS data generation and processing are described elsewhere (Beecham et al, 2017; Leung et al, 2019). In brief, the sample included 498 AD cases and 84 cognitively normal elderly controls from 44 non-Hispanic Caucasian and 67 Caribbean Hispanic families. All studies involved were approved by their respective University Institutional Review Boards (IRBs), and the overall study was approved by the University of Pennsylvania IRB. WGS data were generated using Illumina's 2500 HiSeq platform by the NHGRI's LSACs at the Baylor College of Medicine (BCM), the Broad Institute (BI), and the McDonnell Genome Institute at Washington University (WashU). BCM provided 166 samples with a mean template size of 370 bp (SD = 12.4 bp). For the BI, 232 samples were sequenced with a mean template size of 335 bp (SD = 1.4 bp). WU provided 186 samples with three library preparations targeted at insert sizes of 200, 400, and 550 bp. These three library sizes were chosen to increase SV calling accuracy by incorporating longer reads; however, there was considerable size heterogeneity in the 550-bp read group. Three samples from one family were sequenced at all three LSACs as triplicates for evaluating and adjusting for center-specific sequencing effects.

## Deletion variant calling protocol

Two complementary pipelines for deletion calling, merging, genotyping, and reassembly were implemented (Fig 1). In one approach, each genome was divided into 47 regions (two regions per autosome and one each for X, Y, and mitochondrial chromosomes) excluding telomeres and centromeres and called in parallel using Scalpel (Narzisi et al, 2014) to reduce processing time across the entire genome. Scalpel reassembles gapped alignments using the de Bruijn graph method to increase calling specificity in regions characterized by complex repeat structures. Scalpel was also used to generate precise breakpoints via local assembly within a 1,000-bp capture window for the whole genome. GenomeSTRiP (McCarroll et al, 2006) was used to perform joint genotyping and provide missing genotype information to further refine calls. The second deletion calling pipeline was based on Parliament (English et al, 2015), which created a unified project-level variant call file by combining and filtering calls based on consensus and quality metrics from eight indel/SV callers including Scalpel (Table 1). Parliament also provided gene annotation, genotyping, and local hybrid assembly. Because Parliament's breakpoint detection process is computationally intensive, we limited the analysis to deletions >100 bp. The functional annotation of each variant was determined using SnpEff (Cingolani et al, 2012).

## Sensitivity analysis using simulated spike-in data

We evaluated sensitivity by "spiking-in" SV/indels using BAMSurgeon (Ewing et al, 2015) into triplicated samples (three samples sequenced at all three LSACs). First, we generated a list of predefined SV/indels, including 4,040 deletions and insertions, and 1,560 inversions and tandem duplications, totaling 11,200 events. SV/indels ranged in size from 2 to 5,000 bp and were spiked into all autosomes for the three sample replicates (nine files in total). Half of the spike-in events were inserted as heterozygotes and half as homozygotes. BAMSurgeon failed to add in a small fraction (2.92%) of the attempted spike-in events, and those sites were excluded from sensitivity analysis. For sites where BAMSurgeon succeeded, there were minor discrepancies in the exact breakpoints of the actual spike-in location as compared to its targeted location. These minor breakpoint discrepancies did not affect the results because we applied a 50% reciprocal overlap for detecting spiked-in events. Finally, SV/indels were called for the nine spiked-in samples to measure the sensitivity of each caller across the full range of sizes. Because the true events were known or spiked-in, the sensitivity (Equation (1)) of each SV/indel caller was estimated as follows:

$$Sensitivity = \frac{\#\ of\ Detected\ Events}{\#\ of\ True\ Events} \tag{1}$$

## D-score: a metric for evaluating SV/indel caller specificity in family studies

To ascertain the specificity of deletion calls, we developed the following family-based metric called the deletion or D-score. It represents the log-likelihood ratio of the probability of sharing a

variant by siblings assuming that (a) the variant is true and (b) the variant is a false call. The variant sharing probabilities among siblings depend on the caller sensitivity.

$$D(V) = \log \frac{P_1\left(f_{sib} < f_{sib}^{observed}\right)}{P_0\left(f_{sib} > f_{sib}^{observed}\right)} \qquad (2)$$

$f$ = overall call rate (proportion of samples where a call is made).
$f_{sib}$ = $P_1$(call|call in sib), for a true variant.
For reproducible false calls, $E_0[f_{sib}^{observed}] = f$ = population call rate.
For true variants, $E_1[f_{sib}^{observed}] = f_{sib}$.

$\beta_{HET}$, $\beta_{HOM}$ = the sensitivity of the caller for heterozygous and homozygous variants, specific to each caller and each library design for the sequencing sites. Caller sensitivity vectors were calculated from the spike-in study results.

$$f_{sib} = F(f, \beta_{HET}, \beta_{HOM}) \qquad (3)$$

Thus, given $\beta_{HET}$ and $\beta_{HOM}$ (which can be estimated from the spike-in data), we can compute $f_{sib}$.

True variants have higher sharing across sibs.

$$f_{sib}^{observed} \sim Binomial(n_{sib-pairs}, f_{sib}) \qquad (4)$$

However, false calls are random across samples.

$$f_{sib}^{observed} \sim Binomial(n_{sib-pairs}, f) \qquad (5)$$

For true calls, $f$ (unrelated sharing frequency) is smaller than $f_{sib}$ (sharing frequency among siblings), resulting in larger positive values of the D-score. The D-score metric does not require genotype information and therefore can be used to evaluate caller specificity in the absence of genotyped calls.

### Kinship coefficient

To assess overall call set quality, a kinship coefficient was calculated using KING (Manichaikul et al, 2010) for all sibling pairs with genotype information of SVs/indels. Because a kinship coefficient of 0.25 is expected for the pooled set of heterozygous joint-genotyped calls, departure from this value indicates systematic errors in SV/indel calling. Because multigenerational data are usually not available in family studies of AD, the kinship coefficient has greater utility than a check for Mendelian inconsistencies and is useful for measuring the overall quality of the genotypes.

### Quality control

Many false positives are the result of poor mapping quality between two or more sites and are characterized by excess heterozygosity. Therefore, a Hardy–Weinberg equilibrium $P$-value threshold of $5 \times 10^{-8}$ was applied to filter calls with excess heterozygosity. The BLAST-like Alignment Tool (BLAT) (Kent, 2002) was used to filter deletions with a low predicted mapping quality or that map to many sites (N > 100) in the genome. Finally, deletions with an alternate allele count of less than five were removed from the final call set.

Parliament's consensus and QC strategy proved to be useful in improving call quality by combining call set metrics and applying heuristics to reduce false positives.

### Computational performance of SV/indel callers

Computational performance benchmarks were obtained for the eight SV/indel programs based on the analysis of 20 randomly selected subjects. Performance benchmarks were derived using automated scripts and included total runtime, peak CPU usage, peak memory usage, and processing core hours. All data were processed using an © Amazon's Elastic Cloud 2 (EC2) extra-large instance with © Intel © Xeon 2.4 GHz CPUs. Scalpel benchmarking results were excluded from this analysis because of its extreme computational demands for processing WGS data.

### Laboratory validation of deletion calls

Subsets of Scalpel- and Parliament-derived deletions of different sizes were selected for validation based on three methods: randomly selected events within specified size bins, predicted LOF, and proximity to 74 candidate AD loci with strong genome-wide association signals. These candidate AD loci were curated from GWAS, candidate gene studies, and multiple family-based studies (Goate et al, 1991; Corder et al, 1993; Levy-Lahad et al, 1995; Sherrington et al, 1995; Patel & David, 1997; Lambert et al, 2013a, 2013b; Cruchaga et al, 2013; Beecham et al, 2014; Escott-Price et al, 2014; Jun et al, 2014, 2016, 2017; Logue et al, 2014; Ruiz et al, 2014; Wetzel-Smith et al, 2014; Steinberg et al, 2015; Tosto et al, 2015; Herold et al, 2016; Jakobsdottir et al, 2016; Kohli et al, 2016; Deming et al, 2017; Mez et al, 2017; Sims et al, 2017; Marioni et al, 2018; Zhou et al, 2018; Baker et al, 2019; Jansen et al, 2019; Kunkle et al, 2019; Zhang et al, 2019; Bis et al, 2020). Validation was performed by PCR across the deletion with custom-designed primers followed by Sanger sequencing. For validation, we tested three SV carriers and one non-carrier for each SV. For the Scalpel-derived deletions, the variants were binned by base pair length (2–19, 20–40, 41–60, 61–80, 81–100, and 101–900 bp). The size ranges examined for the Parliament-derived deletions were 101–900, 901–1,000, and 1,001–17,000 bp. The BLAT from the University of California, Santa Cruz Genome Browser (Kent 2002) was used to search and align variant sequences and surrounding sequences to the human reference genome. Because the BLAT had a minimum requirement of 20 bp, sequences smaller than 20 bp were queried by adding flanking sequences upstream and downstream of the test sequence to bring the length up to 20 bp. Both University of California, Santa Cruz HG19 and HG38 reference genomes were queried using the BLAT. In addition, for each deletion, 100-bp sequences flanking the either side of the event were also queried against the BLAT as a contiguous 200-bp sequence (i.e., variant deletion sequence removed). The BLAT alignment allowed for the visualization of the deletion and surrounding sequence in terms of proximity to genes and repeat sequence and facilitated the identification of instances of clear mis-mapping. The sequence surrounding the variants was extracted from HG38 and used for primer design. For variants where a PCR product of ≤1,200 bp was expected (including the variant sequence), primers were designed outside of the breakpoints to amplify across the deletion sequence.

For deletions where the reference allele was too large to be amplified by a 1,200-bp PCR product, a double PCR approach was used. For the first PCR, one primer was designed within the putative deletion sequence, whereas the other primer was placed external to the deletion breakpoint. The samples containing the reference allele and not containing a deletion would yield a product with this PCR. For the second PCR, both primers flanked the putative deletion. Only samples, which contained the deletion, would yield a product for this PCR. The samples from the three individuals reported as heterozygous or homozygous deletions were used for sequence validation, as well as the one control (or reference) sample. When possible, samples from multiple families were used for validation.

## Data Access

BAM files and variant calls on build hg38 of the human genome from the Alzheimer's Disease Sequencing Project (ADSP) are available through the NIA Genetics of Alzheimer's Disease Data Storage Site (NIAGADS), dataset NG00067. ADSP sequencing data aligned to human genome reference hg37 are available through dbGaP (Accession number: phs000572.v8.p4).

## Supplementary Information

## Acknowledgements

The Alzheimer's Disease Sequencing Project (ADSP) data used are available through the NIA Genetics of Alzheimer's Disease Data Storage Site (NIAGADS), dataset NG00067. The ADSP is comprised of two Alzheimer's disease (AD) genetics consortia and three National Human Genome Research Institute (NHGRI)-funded Large-Scale Sequencing and Analysis Centers (LSACs). The two AD genetics consortia are the Alzheimer's Disease Genetics Consortium (ADGC) funded by NIA (U01 AG032984), and the Cohorts for Heart and Aging Research in Genomic Epidemiology (CHARGE) funded by NIA (R01AG033193), the National Heart, Lung, and Blood Institute (NHLBI), other National Institute of Health (NIH) institutes, and other foreign governmental and non-governmental organizations. The discovery phase analysis of sequence data is supported through UF1AG047133 (to GD Schellenberg, LA Farrer, Drs. Pericak-Vance, Mayeux, and Haines); U01AG049505 to Dr. Seshadri; U01AG049506 to Dr. Boerwinkle; U01AG049507 to Dr. Wijsman; and U01AG049508 to Dr. Goate, and the discovery extension phase analysis is supported through U01AG052411 to Dr. Goate, U01AG052410 to Dr. Pericak-Vance, and U01 AG052409 to Drs. Seshadri and Fornage. Sequencing for the follow-up study (FUS) is supported through U01AG057659 (to Drs. Pericak-Vance, Mayeux, and BN Vardarajan) and U01AG062943 (to Drs. Pericak-Vance and Mayeux). Data generation and harmonization in the follow-up phase is supported by U54AG052427 (to GD Schellenberg and L-S Wang). The FUS phase analysis of sequence data is supported through U01AG058589 (to Drs. Destefano, Boerwinkle, De Jager, Fornage, Seshadri, and Wijsman), U01AG058654 (to Drs. Haines, Bush, LA Farrer, Martin, and Pericak-Vance), U01AG058635 (to Dr. Goate), RF1AG058066 (to Drs. Haines, Pericak-Vance, and Scott), RF1AG057519 (to LA Farrer and Dr. Jun), R01AG048927 (to LA Farrer), and RF1AG054074 (to Drs. Pericak-Vance and Beecham). The ADGC cohorts include the following: Adult Changes in Thought (ACT), the Alzheimer's Disease Centers (ADC), the Chicago Health and Aging Project (CHAP), the Memory and Aging Project (MAP), Mayo Clinic (MAYO), Mayo Parkinson's Disease controls, University of Miami, the Multi-Institutional Research in Alzheimer's Genetic Epidemiology Study (MIRAGE), the National Cell Repository for Alzheimer's Disease (NCRAD), the National Institute on Aging Late-Onset Alzheimer's Disease Family Study (NIA-LOAD), the Religious Orders Study (ROS), the Texas Alzheimer's Research and Care Consortium (TARC), Vanderbilt University/Case Western Reserve University (VAN/CWRU), the Washington Heights/Inwood Columbia Aging Project (WHICAP) and the Washington University Sequencing Project (WUSP), the Columbia University Hispanic-Estudio Familiar de Influencia Genetica de Alzheimer (EFIGA), the University of Toronto (UT), and Genetic Differences (GD). The CHARGE cohorts are supported in part by the National Heart, Lung, and Blood Institute (NHLBI) infrastructure grant HL105756 (Psaty) and RC2HL102419 (Boerwinkle), and the neurology working group is supported by the National Institute on Aging (NIA) R01 grant AG033193. The CHARGE cohorts participating in the ADSP include the following: Austrian Stroke Prevention Study (ASPS), ASPS-Family Study, and the Prospective Dementia Registry-Austria (ASPS/PRODEM-Aus), the Atherosclerosis Risk in Communities (ARIC) Study, the Cardiovascular Health Study (CHS), the Erasmus Rucphen Family Study (ERF), the Framingham Heart Study (FHS), and the Rotterdam Study (RS). ASPS is funded by the Austrian Science Fond (FWF) grant numbers P20545-P05 and P13180 and the Medical University of Graz. The ASPS-Fam is funded by the Austrian Science Fund (FWF) Project I904, the EU Joint Programme—Neurodegenerative Disease Research (JPND) in frame of the BRIDGET Project (Austria, Ministry of Science), and the Medical University of Graz and the Steiermärkische Krankenanstaltengesellschaft. PRODEM-Austria is supported by the Austrian Research Promotion Agency (FFG) (Project no. 827462) and by the Austrian National Bank (Anniversary Fund, Project 15435). ARIC research is carried out as a collaborative study supported by NHLBI contracts (HHSN268201100005C, HHSN268201100006C, HHSN268201100007C, HHSN268201100008C, HHSN268201100009C, HHSN268201100010C, HHSN268201100011C, and HHSN268201100012C). Neurocognitive data in ARIC are collected by U01 2U01HL096812, 2U01HL096814, 2U01HL096899, 2U01HL096902, and 2U01HL096917 from the NIH (NHLBI, NINDS, NIA, and NIDCD), and with previous brain MRI examinations funded by R01-HL70825 from the NHLBI. CHS research was supported by contracts HHSN268201200036C, HHSN268200800007C, N01HC55222, N01HC85079, N01HC85080, N01HC85081, N01HC85082, N01HC85083, and N01HC85086, and grants U01HL080295 and U01HL130114 from the NHLBI with additional contribution from the National Institute of Neurological Disorders and Stroke (NINDS). Additional support was provided by R01AG023629, R01AG15928, and R01AG20098 from the NIA. FHS research is supported by NHLBI contracts N01-HC-25195 and HHSN268201500001I. This study was also supported by additional grants from the NIA (R01s AG054076, AG049607, and AG033040) and NINDS (R01-NS017950). The ERF study as a part of EUROSPAN (European Special Populations Research Network) was supported by European Commission FP6 STRP grant number 018947 (LSHG-CT-2006-01947) and also received funding from the European Community's Seventh Framework Programme (FP7/2007-2013)/grant agreement HEALTH-F4-2007-201413 by the European Commission under the program "Quality of Life and Management of the Living Resources" of 5th Framework Programme (no. QLG2-CT-2002-01254). High-throughput analysis of the ERF data was supported by a joint grant from the Netherlands Organization for Scientific Research and the Russian Foundation for Basic Research (NWO-RFBR 047.017.043). The Rotterdam Study is funded by Erasmus Medical Center and Erasmus University, Rotterdam, the Netherlands Organization for Health Research and Development (ZonMw), the Research Institute for Diseases in the Elderly (RIDE), the Ministry of Education, Culture and Science, the Ministry for Health, Welfare and Sports, the European Commission (DG XII), and the municipality of Rotterdam. Genetic datasets are also supported by the Netherlands Organization of Scientific Research NWO Investments (175.010.2005.011, 911-03-012), the Genetic Laboratory of the Department of Internal Medicine, Erasmus MC, the Research Institute for Diseases in the Elderly (014-93-015; RIDE2), and the Netherlands Genomics Initiative (NGI)/Netherlands Organization for Scientific Research (NWO) Netherlands Consortium for Healthy Aging (NCHA), Project 050-060-810. All studies are grateful to their participants, faculty, and staff. The content of these articles is solely the responsibility of the authors and does not necessarily

represent the official views of the National Institutes of Health or the U.S. Department of Health and Human Services. The four LSACs are as follows: the Human Genome Sequencing Center at the Baylor College of Medicine (U54HG003273), the Broad Institute Genome Center (U54HG003067), the American Genome Center at the Uniformed Services University of the Health Sciences (U01AG057659), and the Washington University Genome Institute (U54HG003079). Biological samples and associated phenotypic data used in primary data analyses were stored at Study Investigators Institutions, and at the National Cell Repository for Alzheimer's Disease (NCRAD, U24AG021886) at Indiana University funded by NIA. Associated phenotypic data used in primary and secondary data analyses were provided by Study Investigators, the NIA-funded Alzheimer's Disease Centers (ADCs), and the National Alzheimer's Coordinating Center (NACC, U01AG016976) and the National Institute on Aging Genetics of Alzheimer's Disease Data Storage Site (NIAGADS, U24AG041689) at the University of Pennsylvania, funded by NIA, and at the Database for Genotypes and Phenotypes (dbGaP) funded by NIH. This research was supported in part by the Intramural Research Program of the National Institutes of Health, National Library of Medicine. Contributors to the genetic analysis data included Study Investigators on projects that were individually funded by NIA, and other NIH institutes, and by private U.S. organizations, or foreign governmental or non-governmental organizations. BN Vardarajan was supported by the National Institute on Aging (UF1AG068028) for characterizing complex structural variation in Alzheimer's disease. We thank William J Salerno for running Parliament.

## Author Contributions

JS Malamon: data curation, formal analysis, visualization, and writing—original draft, review, and editing.
JJ Farrell: data curation, software, validation, investigation, visualization, methodology, and writing—original draft, review, and editing.
LC Xia: formal analysis, methodology, and writing—review and editing.
BA Dombroski: formal analysis, validation, investigation, methodology, and writing—original draft, review, and editing.
RG Das: formal analysis and writing—review and editing.
J Way: data curation, formal analysis, and writing—review and editing.
AB Kuzma: data curation, project administration, and writing—review and editing.
O Valladares: software, formal analysis, investigation, visualization, and writing—review and editing.
YY Leung: conceptualization, data curation, software, project administration, and writing—original draft, review, and editing.
AJ Scanlon: formal analysis and writing—review and editing.
IAB Lopez: formal analysis and writing—review and editing.
J Brehony: formal analysis and writing—review and editing.
KC Worley: formal analysis, supervision, funding acquisition, methodology, and writing—original draft, review, and editing.
NR Zhang: data curation, software, formal analysis, supervision, methodology, and writing—original draft.
L-S Wang: data curation, supervision, funding acquisition, and writing—original draft, review, and editing.
LA Farrer: conceptualization, resources, supervision, funding acquisition, project administration, and writing—original draft, review, and editing.
GD Schellenberg: conceptualization, supervision, funding acquisition, methodology, and writing—original draft, review, and editing.
W-P Lee: resources, data curation, formal analysis, supervision, and writing—original draft, review, and editing.
BN Vardarajan: conceptualization, resources, formal analysis, supervision, funding acquisition, project administration, and writing—original draft, review, and editing.

## Conflict of Interest Statement

The authors declare that they have no conflict of interest.

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
