## [Reviewer comments · Life Science Alliance]

Life Science Alliance

A comparative study of structural variant calling in WGS from Alzheimer's Disease families.

John Malamon, John Farrell, Li Xia, Beth Dombroski, Reuben Das, Jessica Way, Amanda Kuzma, Otto Valladares, Yuk Leung, Allison Scanlon, Irving Lopez, Jack Brehony, Kim Worley, Nancy Zhang, Li-San Wang, Lindsay Farrer, Gerard Schellenberg, Wan-Ping Lee, and Badri Vardarajan

DOI: <https://doi.org/10.26508/lsa.202302181>

Corresponding author(s): *Badri Vardarajan, Columbia University Medical Center*

Review Timeline:

Submission Date:	2023-05-24
Editorial Decision:	2023-06-30
Revision Received:	2024-01-22
Editorial Decision:	2024-01-23
Revision Received:	2024-02-07
Accepted:	2024-02-08

Transaction Report:

June 30, 2023

Re: Life Science Alliance manuscript #LSA-2023-02181-T

Badri N Vardarajan
Columbia University

Dear Dr. Vardarajan,

Thank you for submitting your manuscript entitled "A comparative study of structural variant calling strategies using the Alzheimer's Disease Sequencing Project's whole genome family data." to Life Science Alliance. The manuscript was assessed by expert reviewers, whose comments are appended to this letter. We invite you to submit a revised manuscript addressing the Reviewer comments.

Thank you for this interesting contribution to Life Science Alliance. We are looking forward to receiving your revised manuscript.

Sincerely,

B. MANUSCRIPT ORGANIZATION AND FORMATTING:

Reviewer #1 (Comments to the Authors (Required)):

This is a valuable study that tested the existing packages used for deletion calls in human whole genome sequencing data, including their sensitivity and specificity. This work proved that different callers varies considerably in sensitivity depending on the deletion size. SWAN seems the best caller across all sizes based on their comparison. They used D-score to evaluate the specificity. Again, they found the specificity varies depending on the deletion size and sequencing data. They also tested computational requirements and found scalpel and SWAN require more CPU but with high sensitivity and specificity. So Authors picked scalpel to analyze 584 ADSP samples.

- They didn't evaluate Parliament but selected it directly without explanation.
- They performed sanger sequencing to validate 106 deletions. However, they didn't explain the standard for picking deletions for validation. How many subjects did they select for the sanger sequencing? They mentioned a percentage of deletions were tested; what are they according to?
- When "80% of events between 80-100 bp were confirmed by Sanger sequencing", what are the reasons 20% of events were not validated?
- "For randomly selected large events (between 101-900 bp), the validation rate fell to 17%." 17% is low; I wonder if the authors should validate all the deletions.
- They found some AD-associated deletions reported before; what packages did they use? This also indicates that other approaches are reliable in identifying the AD pathogenic variants. Also, this study includes AD and controls; they didn't mention if those published SNVs are associated with AD in this study.

Reviewer #2 (Comments to the Authors (Required)):

Summary:

Drs. Malamon, Vardarajan and colleagues discusses an interesting study comparing the different structural variant (SV) genotyping platforms. Evaluating the commonalities, and challenges and proposing recommendations for calling SVs, particularly the deletions. The rationale of the study is sound and well-detailed in the introduction. However, I think a bit of explanation/rephrasing has to go in the introduction on SVs, making it accessible for more general readers, such as the LSA readership.

Additionally, this reviewer identifies the following points that need to be addressed.

Concerns:

- Methods section mentions including scalpel in creating a unified variant call using parliament. However, Figure 1 does not reflect this strategy. Additional clarification is needed.
- A brief introduction/rationale to spiking-in artificial variants will be very useful in the introduction.
- There seems some contradiction in the size parameters for the parliament-derived deletions (>200bp) and their validation through PCR. As it mentions including deletions ranging from 100bp. Additional clarification is needed.

A comparative study of structural variant calling strategies using the Alzheimer's Disease Sequencing Project's whole genome family data.

Response to Reviewers

We thank the reviewers for their critique of our manuscript. Please find below our point by point response to each question/comment from the reviewers.

Reviewer #1 (Comments to the Authors (Required)):

- They didn't evaluate Parliament but selected it directly without explanation.
Parliament is a consensus structural variant (SV) framework that uses several best-in-class callers to identify high-quality SVs from short-read DNA sequence data at scale. We used Parliament as an alternative to Scalpel/GenomeStrip pipeline to call deletions and evaluated the performance of the two pipelines (Figure 1). We further clarified in the text that we use Parliament as alternative pipeline and benchmarked its performance.

- They performed sanger sequencing to validate 106 deletions. However, they didn't explain the standard for picking deletions for validation. How many subjects did they select for the sanger sequencing? They mentioned a percentage of deletions were tested; what are they according to?
We added text to the manuscript clarifying the rationale for picking the deletions. We previously stated that the three classes of deletions were validated. *"Subsets of Scalpel and Parliament-derived deletions of different sizes were selected for validation based on three methods: randomly selected events within specified size bins, predicted LOF, and proximity to 74 candidate AD loci with strong genome-wide association signals.* We now added the number of carriers and non-carriers that were tested for each SV and the criteria nominate a positive validation.

- When "80% of events between 80-100 bp were confirmed by Sanger sequencing", what are the reasons 20% of events were not validated?
In Supplementary Table 3 we provided details of the SV validation. We declared a SV event to be validated if three SV carriers and one non-carrier was validated by PCR to have the exact genotype as called by the software. For events between 81-100bp called by Scalpel, 8 out of 10 (80%) events were validated. Of the 2 events that we labeled as "failed validation", one was an alternate event. Scalpel called the variant heterozygous in the sample, but the PCR validation identified a homozygous event. The second SV event failed PCR validation.

- "For randomly selected large events (between 101-900 bp), the validation rate fell to 17%." 17% is low; I wonder if the authors should validate all the deletions.

As mentioned in the manuscript, we added further QC for larger deletion calls. Deletion calls mapping to multiple regions of the genome were excluded because they are prone to false positive due to low complexity and lower quality of uniquely mapping to a specific genome location. Using this filter, the validation rate increased to 50% in the genome. We conclude that *"This Sanger sequencing validation of deletions demonstrates that the variants called by Scalpel, particularly within the 2-100 bp size range, are highly reliable and are suitable for genetic association studies."* Validating all deletions is

beyond the scope of this work (we do not have access and consent to DNA of all sequenced individuals from the ADSP cohort).

- They found some AD-associated deletions reported before; what packages did they use? This also indicates that other approaches are reliable in identifying the AD pathogenic variants. Also, this study includes AD and controls; they didn't mention if those published SNVs are associated with AD in this study.

We used two pipelines to detect the deletions as mentioned in the text. After annotation, we used previously published AD GWAS studies to determine SVs in genes near known AD-associated loci. The NHW samples used in this study contributed to large GWAS studies that identified AD association (Kunkle et al 2019, Lambert et al, 2013 and Naj et al, 2011). The GWAS hits were significant in the NHW cohort of this study and showed reduced significance in Caribbean Hispanic cohort.

Reviewer #2 (Comments to the Authors (Required)):

- Methods section mentions including scalpel in creating a unified variant call using parliament. However, Figure 1 does not reflect this strategy. Additional clarification is needed.

We edited figure 1 to include Scalpel as one of the SV callers that was used in Parliament integration.

- A brief introduction/rationale to spiking-in artificial variants will be very useful in the introduction.

We added this text to the introduction of the manuscript.

In addition, we spiked in structural variants in existing sequences for benchmarking and evaluating the performance of several of these tools and pipelines. Spiking in known structural variants allows for a systematic evaluation of a tool's sensitivity and specificity. Spiking in variants of different sizes and complexities allowed us to benchmark tools across a spectrum of genomic alterations, providing a more comprehensive evaluation. Additionally, it also facilitated evaluation of a tool's ability to distinguish true variants from background noise.

- There seems some contradiction in the size parameters for the parliament-derived deletions (>200bp) and their validation through PCR. As it mentions including deletions ranging from 100bp. Additional clarification is needed.

We corrected the typographical error and changed the text to 100bp (from 200bp). The text now says: "Because Parliament's breakpoint detection step is computationally intensive, we limited the analysis to deletions > 100 bp"

January 23, 2024

RE: Life Science Alliance Manuscript #LSA-2023-02181-TR

Dr. Badri N Vardarajan
Columbia University
630 W 168th St
New York, NY 10032

Dear Dr. Vardarajan,

Thank you for submitting your revised manuscript entitled "A comparative study of structural variant calling in WGS from Alzheimer's Disease families.". We would be happy to publish your paper in Life Science Alliance pending final revisions necessary to meet our formatting guidelines.

- please be sure that the authorship listing and order is correct
- please upload all figure files as individual ones, including the supplementary figure files; all figure legends should only appear in the main manuscript file
- please add ORCID ID for the corresponding author -- you should have received instructions on how to do so
- please add the Twitter handle of your host institute/organization as well as your own or/and one of the authors in our system
- please note that titles in the system and on the manuscript file must match
- the abstract should be a single paragraph not exceeding 175 words
- please add your main, supplementary figure, and table legends to the main manuscript text after the references section
- please mark the panels in Figure 4 so that it is clear which panel is A and which is B
- please add an Author Contributions section to your main manuscript text
- please add callouts for Figures 4A-B and 7A-B to your main manuscript text

A. FINAL FILES:

B. MANUSCRIPT ORGANIZATION AND FORMATTING:

Sincerely,

February 8, 2024

RE: Life Science Alliance Manuscript #LSA-2023-02181-TRR

Badri Vardarajan
Columbia University Medical Center
Neurology
630W 168th Street
New York, NY 10032

Dear Dr. Vardarajan,

Thank you for submitting your Resource entitled "A comparative study of structural variant calling in WGS from Alzheimer's Disease families.". It is a pleasure to let you know that your manuscript is now accepted for publication in Life Science Alliance. Congratulations on this interesting work.

DISTRIBUTION OF MATERIALS:

Again, congratulations on a very nice paper. I hope you found the review process to be constructive and are pleased with how the manuscript was handled editorially. We look forward to future exciting submissions from your lab.

Sincerely,
